# Analysis of Structural Boundary Effects of Copper-Coated Films and Their Application to Space Antennas

Xiaotao Zhou [ID], Huanxiao Li and Xiaofei Ma *

Xi'an Institute of Space Radio Technology, China Academy of Space Technology, Xi'an 710100, China
* Correspondence: maxf041600@sina.com

**Abstract:** Copper-coated films are a solution for flexible electronic devices. One of the applications is a flexible-tension film-deployable antenna, which is a large deployable space antenna with broad application prospects. To analyze the possibility of applying coated films to the antenna, surface accuracy evaluation is required. The finite element method (FEM) was used to analyze the surface accuracy of the copper-coated thin-film structures. Both wrinkling and stretching–bending coupling deformation were considered. Simplified models were applied to study factors influencing the surface accuracy under boundary effects. Different sizes of coated area and different boundary conditions were simulated. The results showed the characteristic boundary effects of copper-coated thin-film structures and the influence curve of film thickness and patch size on boundary effects. These findings will inform the design and analysis of variable-stiffness thin-film antennas. On this basis, the application of a flexible-tension film-deployable antenna is discussed, along with a measure to improve the surface accuracy.

**Keywords:** copper-coated films; surface accuracy; wrinkling analysis; variable-stiffness structures; antenna



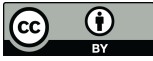

## 1. Introduction

Dielectric-coated films are becoming indispensable carriers in the field of flexible electronics and are widely studied in the fields of wearable devices and aerospace antennas [1]. Thin-film phased-array antennas are a new generation of phased-array antennas based on flat-panel phased-array antennas, combined with thin-film composite materials as carriers [2]. The direct-array flexible thin-film antenna is an active antenna where the reflecting surface made of the T/R (transmitter/receiver) module is integrated on the film or on the deployable structure to perform the function of the antenna, as shown in Figure 1 [3–5].

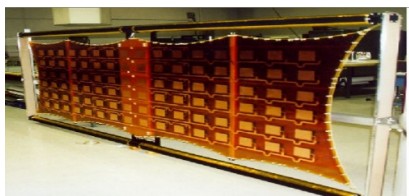

**Figure 1.** Planar phased-array film antenna by JPL.

Although the advantages of the thin-film antenna are remarkable, such as a light weight and little gathered volume, maintaining the profile accuracy of the antenna is a major challenge. Uneven loading, inaccurate assembly, and assembly and transport processes can lead to varying degrees of film wrinkles [6]. The main effects of folds on thin-film antennas are as follows: the presence of smaller folds leads to diffuse reflections and small-amplitude performance degradation; localized wrinkles formed by larger folds will lead to the appearance of hot spots, which may exceed the maximum failure temperature

and lead to material failure [7]. In addition, a phased-array antenna always consists of multiple layers, including a radiating layer and a reflecting layer. The distance between the two layers is determined by the electric magnetic performance. A small deformation of the distance may cause large changes in electric magnetic performance, since phased-array antennas are equivalent to multiple antennas superimposing electromagnetic signals in space according to their position. Hence, the research around the design and analysis of thin-film antennas is a hotspot [8–14].

The current method of design and analysis of wrinkles is mainly based on the tension field theory and thin plate stability theory [15–28]. Tension field theory ignores the bending strain energy of thin-film structures and can simply and effectively determine the critical state, region, direction, and surface stress or strain of thin-film folds, but it cannot obtain detailed information on wrinkling waves, nor can it obtain the specific evolution law of wrinkles. In contrast, the thin plate stability theory considers that thin-film structures have structural bending strain energy; it uses the compressive stress reaching the critical buckling load as the basis for determining instability, and it can obtain specific characteristic parameters of folds and their evolution laws, which is the main theoretical approach used for the study of thin-film instability mechanisms. Published works have mainly analyzed the wrinkling behavior of pure membrane structures, but the space-tensioned thin-film phased-array antenna is a variable-stiffness film structure, and its surface accuracy is limited to within a small fluctuation range [29]. In recent years, several analyses of variable-stiffness film structures have been conducted [30,31]. Yan et al. [32] analyzed the characteristics of wrinkles related to the positions of rigid elements and the underlying wrinkling mechanism of a membrane with square rigid elements. Qiu et al. [33] proposed a data-driven framework for the wrinkling design of inhomogeneous sheets by a trained backpropagation algorithm. Variable-stiffness film structures could be seen as thin-walled laminate structures. Buckling, post-buckling, and wrinkling analyses of thin-walled laminate structures are widely researched [34–42]. Jia et al. [38] explored the design space for the nonlinear buckling of composite thin-walled lenticular tubes under pure bending. Lewicka M. et al. [39] provided a derivation of the von Kármán equations for the shape of and stresses in an elastic plate with incompatible or residual strains. Compared to homogeneous thin-film structures and thin-walled laminate structures, variable-stiffness film structures have been less well studied [40,41].

The structural design for the analysis of tensioned membrane antennas has been researched in terms of space environment loads and experiments [42–46]. However, surface accuracy considering stiffness variations has not been researched. In published works, an analysis of deformation coupling factors due to coatings in terms of variable-stiffness thin-film structures has not yet been seen. Since large performance changes are possible with small variations in surface accuracy, focusing on out-of-plane deformation analysis and surface accuracy maintenance of variable-stiffness film structures is necessary to evaluate the applicability of space-tensioned thin-film phased-array antennas.

In this paper, boundary effects and the influence of the thickness of coated thin-film structures are analyzed. The coated thin-film structure was considered to be a variable-stiffness film structure, and the coated areas (patches) were regarded as layups rather than rigid boundary conditions. A wrinkling analysis was carried out by the finite element method, and two simplified models were used to analyze the influence of the size of patches and the thickness of the membrane on the surface accuracy. Three typical modes of induced wrinkling deformation under the influence of boundary effects were analyzed. The results show the variation rule of the out-of-plane deformation characteristics of variable-stiffness thin-film structures with boundary effects and differing film thickness. An explanation of the deformation mechanism is discussed. The results are helpful for the design of a space-tensioned thin-film phased-array antenna. The application to space antenna by a coated thin-film structure is discussed in terms of an improved surface accuracy method.

## 2. Materials and Methods

### 2.1. Geometry and Simplified Model

Figure 2 shows the structural geometry of a double-layered space-tensioned thin-film antenna, which is a typical coated thin-film structure. The yellow pieces represent patched areas, which actually are thin copper foil. The membrane area is made of thin Kapton or rubber composite film. Based on former studies, the boundary shape of the membrane is parabolic.

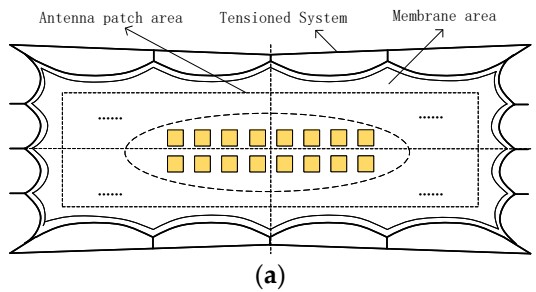 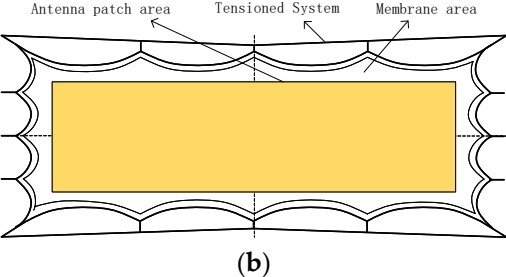

(**a**)                                                             (**b**)

**Figure 2.** A schematic diagram of the space-tensioned thin-film antenna structure. (**a**) Radiation layer; (**b**) grounding layer.

To analyze the influence of the tensioned system on structural deformation, simplified models were applied. The wrinkling area always exists in areas close to the concentrated load, so three typical models were chosen:

1.  Model 1: Corner-tensioned variable-stiffness model (Figure 3);
2.  Model 2: Different boundary shape models (dotted line in Figure 3).

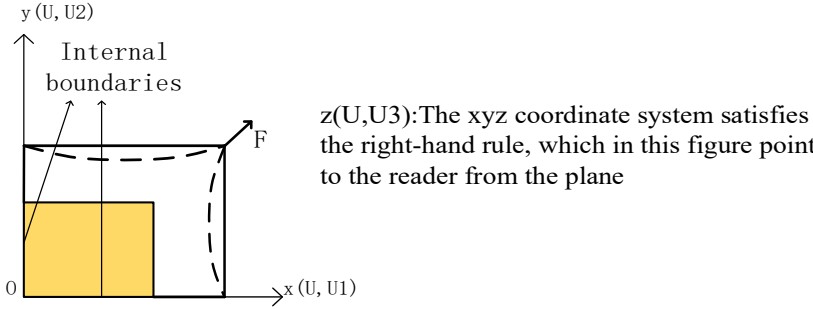

**Figure 3.** Simplified models: Corner-tensioned variable-stiffness model (with simulation coordinate system Oxyz statement).

The models were built by ABAQUS as finite element models (FEMs). The boundary setting of model 1 was two inner sides setting symmetrical and two outer sides setting free with corner point setting displacement. The coordinate system in the simulation is shown in Figure 3, and the coordinate unit is unified as mm. In this paper, the out-of-plane displacement (z-direction displacements, shown as U, U3 in the simulation result plots later) is an important parameter for evaluating the wrinkling deformation.

To analyze the factors influencing electromagnetic performance, the thickness of the membrane and the size of the coated areas were considered. The structural deformation of the antenna was considered both wrinkling deformation and stretching–bending coupling deformation. A Wrinkling deformation is widely seen in thin-film structures. A stretching–bending coupling deformation is led by asymmetrical layers in patched areas because of asymmetrical layers leading to variation in bending stiffness along the pavement direction.

### 2.2. Finite Element Model and Simulation Method

The structural deformation on thin-film antenna was analyzed by ABAQUS 2020 with python scripts suitable for parametric studies. Four-node general-purpose shell elements with reduced integration (ABAQUS element type S4R) were used.

The size of model 1 was $500 \times 200$ mm, tension displacement was 0.01 mm. The size of model 2 was $100 \times 100$ mm, tension displacement was 0.2 mm (the tensile force required in space is not large and is within the elastic range). The material parameters are as follows:

- Membrane: Kapton or rubber composites, Young's modulus: 6 MPa, Poisson's ratio: 0.3, initial thickness: 0.025 mm.
- Coated areas: copper foil, Young's modulus: 110 GPa, Poisson's ratio: 0.33, thickness: 0.005 mm.

The area membrane with patches was modeled by composite layups. The asymmetries along the layer orientation will lead to stretching–bending coupling deformation. The area without patches was faced with wrinkling deformation. The wrinkling deformation analysis used the post-buckling analysis method of membrane structure [23,47], which offers good approximation accuracy under comparison with analytical model and experiments. For convergence calculations, mesh density was set at 0.4%. The procedure could be divided into two steps:

1. Nonlinear eigenvalue buckling analysis. In this step, initial prestress conditions need to be set for computational convergence. The loading condition was set same as the static analysis process. Solver was set as Lanczos method.
2. Post-buckling analysis. In this step, imperfection was considered. It is chosen from the results in step 1. In this paper, the first six eigenvalue modes were chosen and the scaling factor was set at 1/5 thickness.

As the wrinkling analysis is based on static analysis, stretching–bending coupling deformation [48] and wrinkling deformation are not linear stacked, further discussion will take place in Sections 3 and 4.

## 3. Results and Discussion

### 3.1. Boundary Effects on the Wrinkle Pattern

Figure 4 shows the wrinkling form of the film section when the distance between the coated area and the boundary is varied. Figure 4a shows a single-peak wrinkle. Figure 4b–e shows single-peak, double-valley wrinkles. Figure 4f shows single-peak, double-valley wrinkles and several shorter wrinkles. For descriptive convenience, the width ratio of the patch region to the film region is defined as $\alpha$.

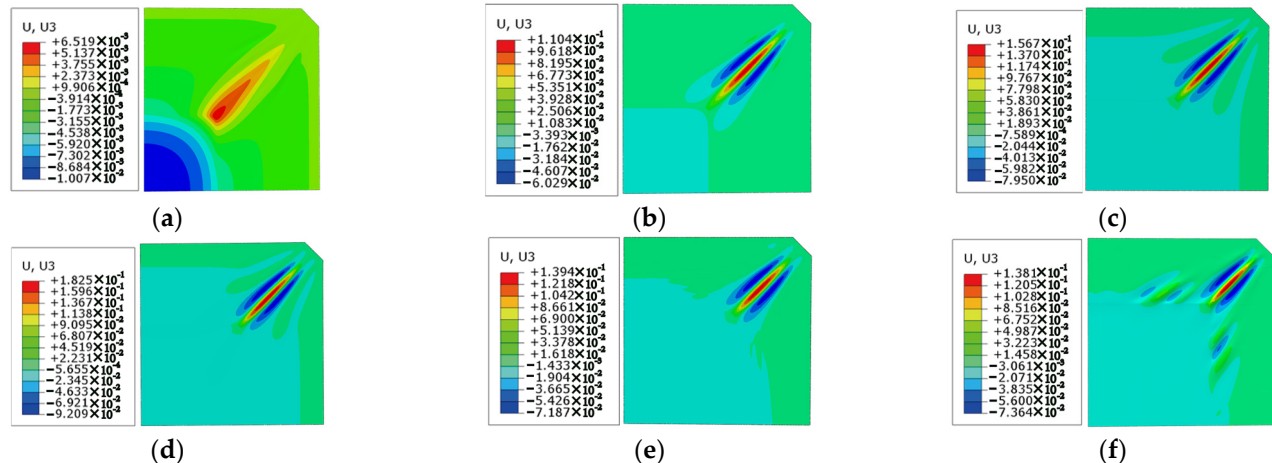

**Figure 4.** FEM wrinkling results of corner loading in different sizes of coated areas (0.025 mm thick membrane, unit of U3 is mm). (**a**) $\alpha = 0.4$; (**b**) $\alpha = 0.45$; (**c**) $\alpha = 0.5$; (**d**) $\alpha = 0.55$; (**e**) $\alpha = 0.6$; (**f**) $\alpha = 0.65$.



Figure 4a shows the curvature balance relationship along the load direction, compared with another five occasions; when the coating area is far from the boundary, deformation occurs only in the direction of the tensile load and only in the axial tension area. Figure 4b–f shows single-peak, double-valley wrinkles caused by the bending moment and stretching-induced buckling behavior. They are due to the curvature balance relationship perpendicular to the load direction. Shorter wrinkles in Figure 4f appear on the boundary of the coated area and point towards the application area, which means that it is caused by shear loading. No shear-induced wrinkles in Figure 4a–e means that shear-induced wrinkles only occur when the aspect ratio of the sheared area is within a certain range (narrow enough).

Figure 5 shows the deformation under corner-tensioned loading. All four pictures show wrinkling patterns in the area between the corner loading area and the coated area. In these situations, compared to Figure 5a,c (or Figure 5b,d), the amplitude of out-of-plane deformation is reduced by applying a parabolic boundary. However, it is different from the pure membrane structure and the wrinkling behavior does not disappear totally. The phenomenon could be explained by the variable stiffness revolved by patches, the variable stiffness led to stress concentration distribution in the inner region of the membrane. Since the boundary condition influences stress distribution caused by boundary loading, the wrinkling could be hardly inhibited totally by applying parabolic boundary conditions.

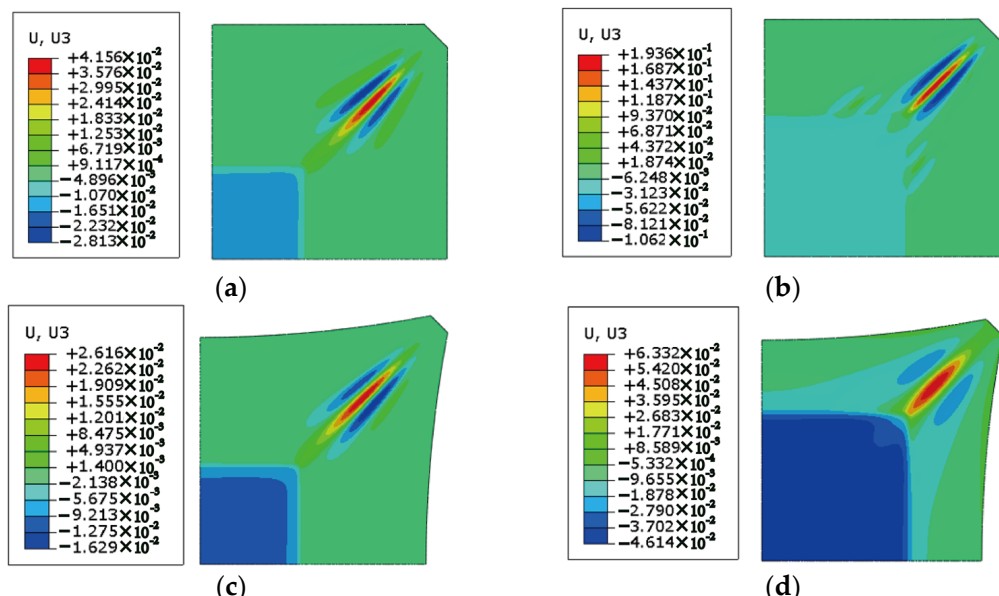

**Figure 5.** FEM wrinkling results of corner loading in different size of patch and thickness of membrane (unit of U3 is mm). (**a**) $\alpha = 0.4$ and rectangular boundary shape; (**b**) $\alpha = 0.6$ and rectangular boundary shape; (**c**) $\alpha = 0.4$ and parabolic boundary shape; (**d**) $\alpha = 0.6$ and parabolic boundary shape.

### 3.2. Effect of Structural Parameters on the Wrinkling Characteristics

Figures 4 and 5 show that neither the size of the coating area nor the boundary shape effects affect the crease deformation characteristics in the direction of the tensile load. Therefore, this section measures the wrinkling characteristics for different structural parameters, with the wrinkling deformation shown in the area of Figure 6a. For the convenience of description, the wavelength in this paper is defined as the width of the wrinkle (on the line of Figure 6), which is a main characteristic of wrinkles.

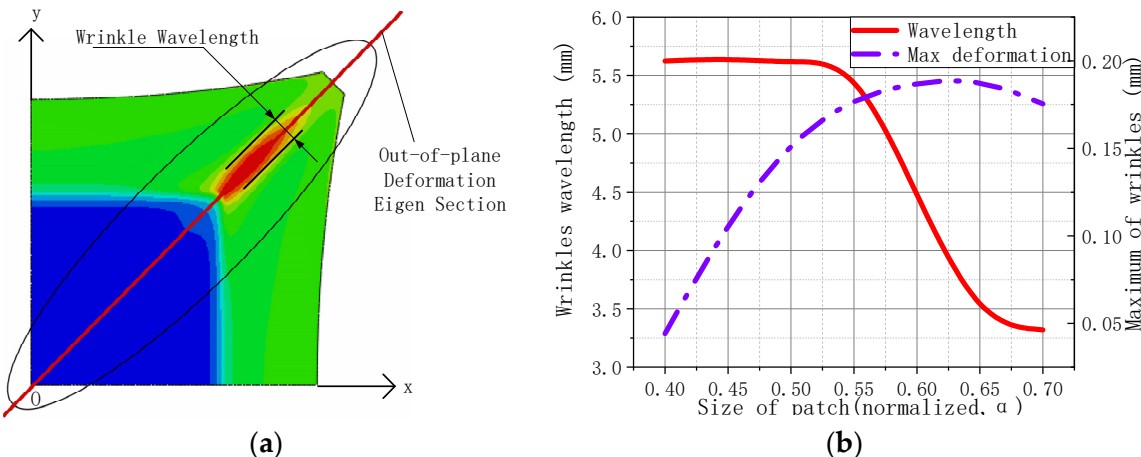

**(a)**         **(b)**

**Figure 6.** Deformation analysis area and patch shape influence curve. (**a**) Deformation analysis region; (**b**) wrinkling wavelengths and out-of-face deformations varies by size of patches.

Figure 6 shows the out-of-plane deformation eigen section and patch shape influence curve. Figure 6b shows that the wavelength and maximum deformation varies by size of patches. The wavelength varies with the size of the variable stiffness area, showing a pattern of gentle variation, then decreasing and then leveling off. When the variable stiffness area size is less than 0.5, the wavelength remains constant at 5.6 mm; when the variable stiffness area size is larger than 0.5 and smaller than 0.65, the wavelength shows a nearly linear decrease from 5.6 mm to 3.5 mm; when the variable stiffness area size is larger than 0.65, the wavelength tends to be stable.

The change in wrinkle extremes shows a tendency to increase and then decrease. The extreme value of the maximum appears at the variable stiffness area size of 0.6–0.65, and the extreme value of the maximum reaches 0.2 mm; the maximum changes from 0.4 mm to 0.5 mm at the variable stiffness area size, and the maximum shows a nearly linear change with the increase in the variable stiffness area size.

Figure 7 illustrates that wrinkling wavelength and max deformation varies by total elongation. Figure 7a shows that as the total elongation increases, there is an overall decreasing trend in the wavelength. The wavelength of the pure film structure remains almost unchanged at a total elongation of less than 0.2%, differing significantly from the fold characteristics of the variable stiffness film structure. And the wavelength decreases by the increasing of the patch size, but these patterns do not behave as a liner relationship.

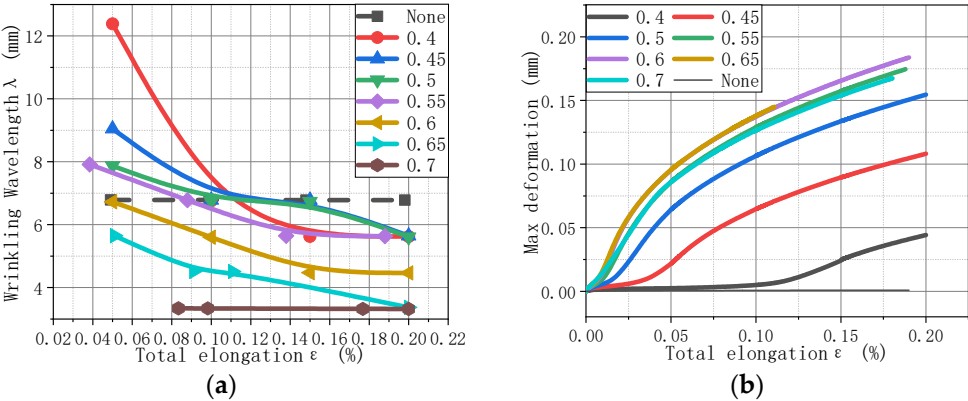

**(a)**         **(b)**

**Figure 7.** Wavelength and max deformation vary by total elongation, (**a**) wavelength (mm) varies by total elongation (%); (**b**) max deformation (mm) varies by total elongation (%).

Figure 7b gives the curve of the deformation extremes with total elongation. When the variable stiffness zone size is greater than 0.5, the deformation poles grow faster at a

total elongation of less than 0.025% and slower at a total elongation of greater than 0.025%. When the variable stiffness zone size is 0.4 and 0.45, a two-part pattern also emerges for the wrinkling polarity trend, but the wrinkling polarity growth trend from the total elongation increase occurs later than in the case of larger variable stiffness zone sizes. Comparing the left and right plots in Figure 7, it can be seen that the deformation poles of the variable stiffness film structure are 2–3 orders of magnitude larger than those of the pure film structure, with the smaller the size of the variable stiffness area, the more the deformation poles with total elongation curves is close to the wrinkling poles with total elongation curves of the pure film structure.

Figure 8 gives a comparison of the out-of-plane fold deformation curves and magnitudes for three thicknesses, two boundaries, and two patch sizes. The thicknesses are the same in each plot, red and orange represent square boundaries while blue and black represent parabolic boundaries, and the horizontal axis can be read off for different width ratios $\alpha$. There are two typical deformation curves in Figure 8, the difference between them is whether the deformation in the film region has a "peak curve" or not.

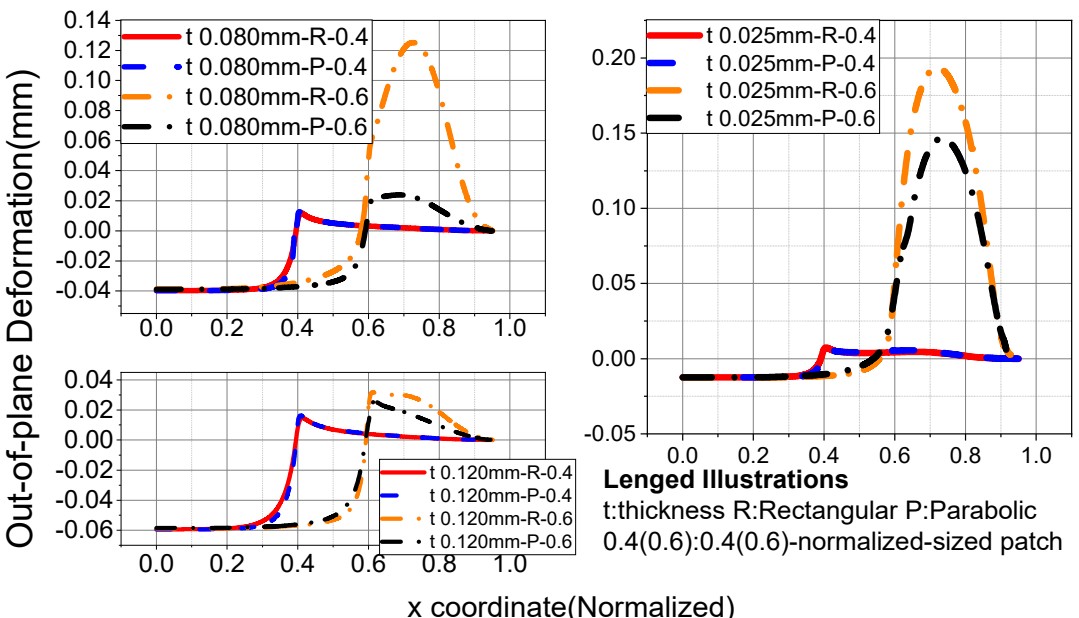

**Figure 8.** Deformation curves at out-of-plane deformation eigen section (Figure 7a) with different thickness, boundary condition and $\alpha$. (The horizontal axis is normalized x-coordinate in Figure 7).

The right panel shows the deformation for the 0.025 mm thick film. The extreme values appear at the patch-film junction area when $\alpha = 0.4$, with a gradual decrease in the partial deformation from 0.4 to 1 normalized $x$ coordinate. The deformation curve when $\alpha = 0.6$ has a sinusoidal-like curve shape. The extreme values corresponding to the parabolic boundary are about 3/4 of the extreme values for the square boundary. The upper left panel shows the deformation pattern at 0.080 mm thickness. The deformed shape is sinusoidal-like shape when $\alpha = 0.6$ and is a square boundary case. However, the deformation pattern for the parabolic boundary case when $\alpha = 0.6$ shows similar patterns with the deformation pattern when $\alpha = 0.4$. The extreme value of the three conditions appeared at the patch-film junction area. As can be seen in the lower left panel, at 0.120 mm thickness the deformation patterns are similar for all cases, with the extremes occurring at the patch-film junction area. The comparison of the three graphs show that it would be easier for a sinusoidal deformation curve with $\alpha$ increasing and the thickness of the film decreasing.

It could be concluded that with the increase in membrane stiffness, the wrinkling performance contribution to structural deformation would decrease, and the main deformation shape would be caused by the stretching–bending coupling effect.

Figure 9 shows deformation at the out-of-plane deformation eigen section with different thicknesses of film. $\alpha = 0.4$ in the upper two figures, while $\alpha = 0.6$ in the lower two pictures. The left two graphs are obtained by the structure with rectangular boundaries, while the right two are with parabolic boundaries.

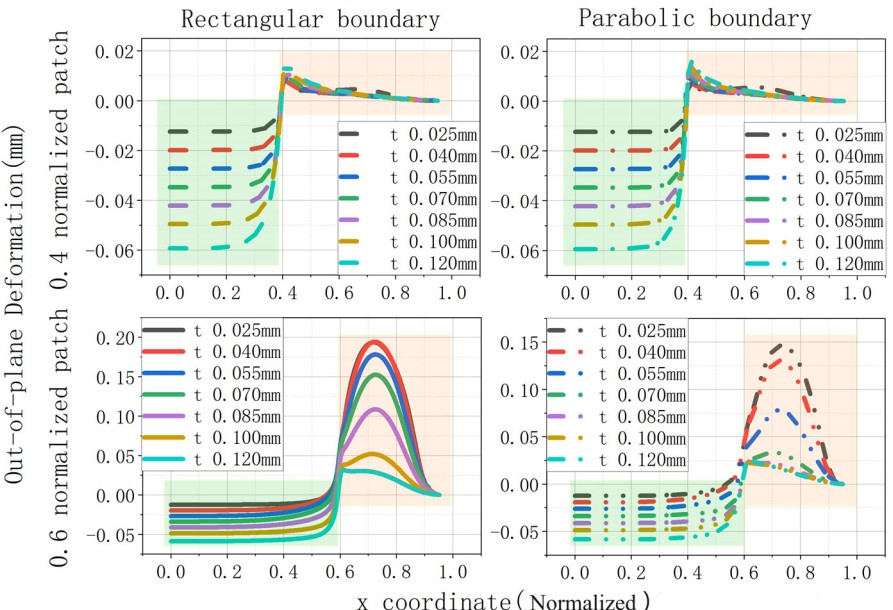

**Figure 9.** Deformation curves of different thickness at out-of-plane deformation eigen section under various boundary conditions and $\alpha$ (the light green region represents coated areas and the light orange region represents pure film areas). (The horizontal axis is normalized x-coordinate in Figure 7).

The maximum curves at $\alpha = 0.4$ appeared at the patch-film junction area at all thicknesses and both boundary conditions. As the thickness increases, the deformation of the patched area increases steadily (absolute values). And the value of deformation in the patched area is greater than that in the membrane area (absolute values). The extremes of the deformation curves at $\alpha = 0.6$ show different patterns than at $\alpha = 0.4$. The maximum appeared in the film region when the thickness is small. Specifically, when the thickness is smaller than 0.1 mm in rectangular boundary conditions, the maximum is about 0.72 normalized x coordinate. When the thickness is smaller than 0.07 mm in the parabolic boundary, the maximum is about 0.72 normalized x coordinate. The absolute values of the minimum values are smaller than the absolute values of the maximum values when the thickness is smaller than 0.85 mm in a rectangular shape and 0.07 mm in a parabolic shape. Comparing the results for different boundaries, it can be seen that there is little difference in the deformation due to the boundaries at $\alpha = 0.4$. At $\alpha = 0.6$, the deformation of the film region with parabolic boundaries decreases much faster with the increase in thickness than the deformation of the film region with square boundaries. At $\alpha = 0.6$, the sinusoidal-like deformation in the film region starts to disappear when the film thickness is above 0.1 mm for the rectangular boundary condition, and the sinusoidal-like deformation in the film region starts to disappear when the film thickness is above 0.07 mm for the parabolic boundary. Moreover, the deformation extreme value of the parabolic boundary is more than 25% smaller than that of the rectangular boundary. The parabolic boundary can significantly change the in-plane stress distribution near the boundary [5] and reduce the shear flow near the boundary, which is the reason for its ability to suppress tensile-induced wrinkles.

To conclude, the wrinkling patterns appear more easily in rectangular boundaries and at $\alpha = 0.6$.

### 3.3. Discussion of Typical Deformation Curves

In order to seek theoretical support for the phenomena in Figures 4–9 for the purpose of informing engineering design, two qualitative analyses are conducted in this section.

(1)    Stretching–bending coupling effect analysis

Deformation patterns can be analyzed by stress analysis and the thin plate deformation theory. Figure 10 gives an illustration of the stress distribution. Figure 10a shows the stress distribution at the corner tensioning of the concentrated load, and it is worth noting that in this case the distribution is uniform in the circumferential direction. Figure 10b shows the stress distribution for a localized uniform load, which is in the form of an axial tensile load extending to the inner section of the film due to the presence of a section of uniform load, but outside this area (near the free boundary), the stresses are still uniformly distributed in a circumferential direction. Based on this, Figure 10d shows the superposition of stress states. The squares with black borders represent coated areas, which have very little displacement at its boundary due to the much greater tensile stiffness of the coated area than the film. But, due to the asymmetrical lay-up arrangement, a bending load is generated at coated area's boundary, which is caused by the transfer of tension $F$ to the connection between the coated area and the film, and the force $f$ perpendicular to the coated area represents this equivalent load; $f$ is not on the middle surface of the coating area and is conducted by the composite interface between the film and the coated area. It is worth noting that $f$ is not perfectly uniformly distributed over the boundary of the coating area, but is related to the distance of $F$.

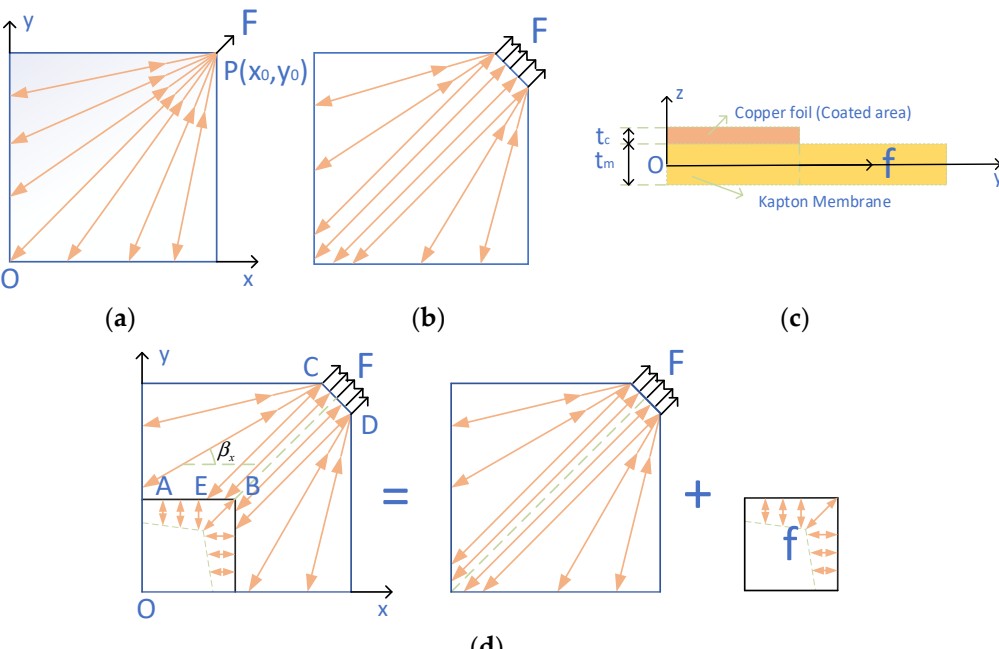

**Figure 10.** Stress analysis of angle point tensioned films and variable stiffness films. (**a**) Concentrated load tensioning stress distribution; (**b**) localized uniform load tension stress distribution; (**c**) illustration of thickness direction distribution; (**d**) stress decomposition of variable stiffness film structures.

The derivation process is as follows:
Suppose that any point on $AB$ is $(x, y_{AB})$, then its distance from point $P$ is:

$$l = \sqrt{(x_0 - x)^2 + (y_0 - y_{AB})^2},\ x \in (0, x_B) \tag{1}$$

Suppose the length of $CD$ is $l_{CD}$, then the uniform loading could be expressed as $F/l_{CD}$. The internal forces in the circumferential load area are:

$$f_{\sigma-mem} = \frac{F}{l_{CD}} \Big/ \frac{\pi}{2} = \frac{2F}{\pi l_{CD}} \tag{2}$$

The axial tension and tangential shear forces on the boundary of the coated area are:

$$\begin{cases} f_{\sigma-co} = f_{\sigma-mem} \sin\beta_x = \frac{2F}{\pi l_{CD}} \frac{(y_0 - y_{AB})}{l} \\ f_{s-co} = f_{\sigma-mem} \cos\beta_x = \frac{2F}{\pi l_{CD}} \frac{(x_0 - x)}{l} \end{cases} \tag{3}$$

$\beta_x$ represents the angle between the line from any point on $AB$ to the point $P$ and the x-axis (Figure 10a). As the film is much weaker than the coating area in terms of stiffness, the applied load has less effect on the in-plane deformation of the coating area. Therefore, the tangential deformation under $f_{s-co}$ of the coating area is negligible in this derivation process.

Combined with Figure 10c, the bending moment at any point on $AB$ is:

$$M_{co} = f_{\sigma-co} \times \frac{t_c + t_m}{2} = \frac{F}{\pi l_{CD}} \frac{(y_0 - y_{AB})(t_c + t_m)}{\sqrt{(x_0 - x)^2 + (y_0 - y_{AB})^2}} \tag{4}$$

The only variable in Equation (4) is $x$, so the extreme value of the bending moment occurs at point B (For the load forms in Figure 6b, the bending moments on $BE$ are all maximum).

So, the wrinkling deformation region in the form of the deformation shown in Figures 4 and 5 is influenced by the stretching–bending coupling effect.

Considering the thin plate theory, then:

$$\begin{cases} M_x = D(\chi_x + \mu\chi_y) \\ M_y = D(\chi_y + \mu\chi_x) \\ M_{xy} = M_{yx} = D(1 - \mu)\chi_{xy} \end{cases} \tag{5}$$

In the above equation, $D = \frac{Et^3}{12(1-v^2)}$, $\chi_x = \frac{\partial^2 \omega}{\partial x^2}$, $\chi_y = \frac{\partial^2 \omega}{\partial y^2}$, $\chi_{xy} = \frac{\partial^2 \omega}{\partial x \partial y}$ $\omega$ represents out-of-plane deformation.

The bending moment at any point on the $AB$ side could be regarded as $M_x$ in Equation (5). The bending moment on the other side is xy-axis symmetric to $M_{co}$.

Combining Equations (4) and (5), then:

$$(\chi_x + \mu\chi_y)_{AB-i} = \frac{M_{CO}}{D_i}, (i = membrane, coated\_areas) \tag{6}$$

The bending moment on the $AB$ side is the same for the coating area and the film area. So, the curvature of the coating area and the film area is determined by bending stiffness $D$.

Equation (6) could be used to explain the phenomenon in Figure 10. With the thickness increasing, the bending stiffness $D$ is increasing. So, the deformation in the coated area and membrane area is increasing in Figure 10.

The stiffness of the coated region can be expressed as:

$$D_c = \sum \int_{z_{i-1}}^{z_i} E_i z_i^2 dz = \frac{1}{12(1 - v^2)} E_m t_m^3 + \frac{1}{3} E_c \left( (t_m + t_c)^3 - t_m^3 \right) \tag{7}$$

Assuming that tm is the variable and the other values are the values of the structural parameters of the paper, the order of magnitude of the two components in Equation (7) is analyzed by the following process:

$$\frac{1}{12(1-\nu^2)}E_m t_m^3 \gg \frac{1}{3}E_c\left((t_m + t_c)^3 - t_m^3\right) \tag{8}$$

Simplifying Equation (8), there is:

$$\frac{t_c}{t_m} \ll \sqrt[3]{\frac{E_m}{4E_c} + 1} - 1 \tag{9}$$

Substituting the material data parameters of this paper to Equation (9):

$$t_m \gg \begin{cases} 1104 \ \mu m & D_c = D_m \\ 11041 \ \mu m & 10D_c = D_m \\ 110.4 \ \mu m & D_c = 10D_m \end{cases} \tag{10}$$

From Equation (10), it can be seen that the stiffness of the coating region is 10 times larger than the stiffness of the film region, even if the maximum value of the film thickness of 120 μm in this paper is chosen, so the stiffness of the coating region can be considered to be dominated by the copper coating, and a simplified expression $D_c = \frac{1}{12(1-\nu^2)}E_c t_c^3$ will be used in the subsequent calculations.

The bending stiffness $D$ in the coated region can be considered to be fully provided by the copper coating, so according to Equation (6), the bending deformation in the coated region is independently influenced by $M_{co}$, which is shown in Equation (4) to be a one-time linear function of the film thickness $t_m$, so that Equation (11) could be obtained:

$$\left(\chi_x + \mu\chi_y\right)_{AB-coated\_areas} \sim \frac{(t_m + t_c)}{E_c t_c^3} \sim t_m \qquad D_c \gg D_m \tag{11}$$

Equation (11) shows that the coated region deformation is also a one-time linear function of $t_m$, so the curve intervals in the light green region in Figure 9 exhibit equally spaced sets of plots with thickness.

(2)   Out-of-plane deformation analysis of thin film regions

The coated region can be viewed as being affected only by the stretching–bending coupling effect, whereas the out-of-plane deformation in the film region is affected by a combination of the stretching–bending coupling effect and the buckling effect.

The light orange regions of the upper two panels of Figure 9 exhibit the pure bending behavior of the film structure, while the bottom two plots show wrinkling deformation (caused by buckling). The buckling critical load is now analyzed to illustrate the condition of wrinkling deformation. The film region wrinkles in this paper are buckling due to tensile loading, so the force perpendicular to the Poisson contraction in the $F$ direction can be expressed as:

$$F_p = -\nu F \tag{12}$$

According to the thin plate theory, the critical buckling load of the thin-film structure under compressive loading could be estimated by simply a supporting sheet [49]:

$$F_c = k\frac{\pi^2 D}{b^2} = \left(\frac{mb}{a} + \frac{1}{\frac{mb}{a}}\right)^2 \frac{\pi^2 D}{b^2} \tag{13}$$

In Equation (13), $m = 1, 2, 3, \ldots$, $b$ represents the width of the plate, $a$ represents the length of the plate, $D$ represents the stiffness of the plate.

Combining Equations (12) and (13) and noticing geometric relationship between the length and width of the wrinkling region of the film in Figure 10, the equation could be obtained:

$$vF = F_c = \left( m\frac{l_{cd}}{\sqrt{2}(1-\alpha)l} + \frac{1}{m\frac{l_{cd}}{\sqrt{2}(1-\alpha)l}} \right)^2 \frac{\pi^2}{b^2} \frac{E_m t_m^3}{12(1-\nu^2)} \tag{14}$$

Equation (14) is the condition for wrinkle deformation in the lower two figures in Figure 9. As the thickness increases, the critical value of the buckling load increases, so the fold deformation curves in the lower two plots of Figure 9 become less pronounced as the thickness increases. Based on the method of taking the value of $k$ [49], as $\alpha$ increases, the value of $k$ decreases, so the $F_c$ decreases and stretch-induced wrinkles are more likely to occur.

## 4. Application on Space Tensioned Thin-Film Antenna

Reducing whole stress could be an efficient way to reduce wrinkles. Since the antenna was designed to deploy in space, there would be almost no disturbing force; even if the tensioned forces were balanced partially, the profile shape would keep static balanced. The diagram of this method is shown in Figure 11. This method could be named as the partial compensation tensioned method. At the local extent of the boundary $F_0$, $F_1$ and $F_2$ are in equilibrium with each other so that inside the film, the membrane structure is in a low-stress state and the Poisson effect is not obvious, so the effect of wrinkles in low-stress areas is reduced. At the same time, a low-stress state would reduce the out-of-plane deformation caused by the stretching–bending coupling effect, so the patched area could maintain an asymmetric layups state.

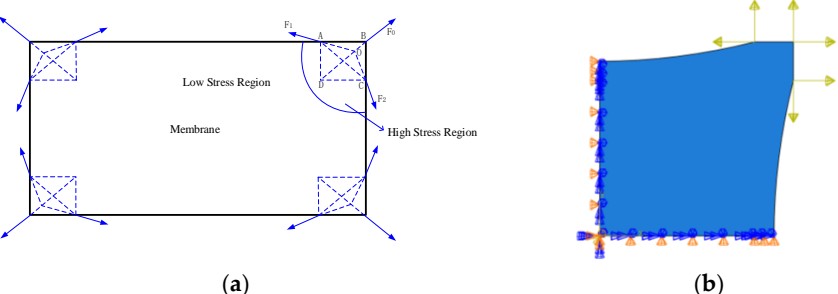

(**a**)　　　　　　　　　　　　　　　　　(**b**)

**Figure 11.** Diagram of partial compensation tension and FEM (the arrows on the diagram all represent forces, as expressed in the upper right-hand corner of the left diagram). (**a**) Diagram; (**b**) FEM (1/4 model).

Figure 12 shows the deformation before and after partial tension is compensated. Although the maximum and minimum values of out-of-plane deformation change little, the deformation area reduced.

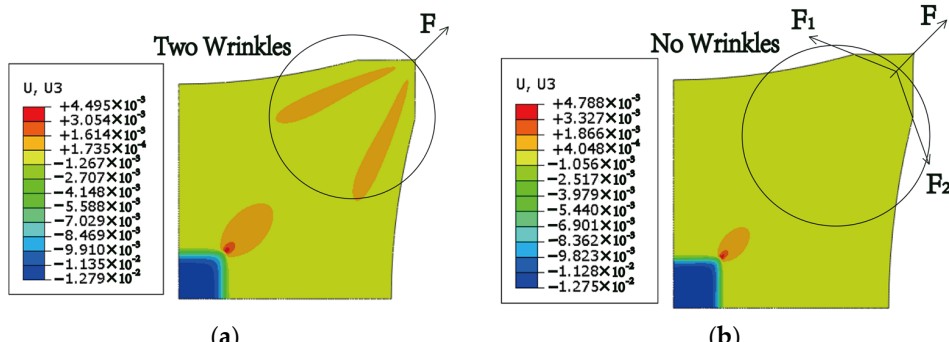

(**a**)　　　　　　　　　　　　　　　　　(**b**)

**Figure 12.** Deformation before and after compensation tensioned (unit of U3 is mm). (**a**) Deformation before compensation tensioned. (**b**) Deformation after compensation tensioned (20% compensation forces).

## 5. Conclusions

The structural boundary effects of surface coated thin-film structures have a multi-modal superimposed effect on their planimetric accuracy. It could be seen as a variable stiffness structure and applied to a space-tensioned thin-film phased-array antenna. Based on the wrinkling analysis in this paper, the boundary effect and thickness of membrane were analyzed in terms of surface accuracy. The simulation results revealed the out-of-plane deformation mechanism of variable stiffness thin-film structures. The conclusions could be drawn as follows:

1. The structural boundary effects of surface coated thin-film structures are characterized by three typical modes of wrinkling deformation, namely bending-induced wrinkling deformation, stretching-induced wrinkling deformation and shear-induced wrinkling deformation. They occur sequentially as the coating area approaches the boundary.

2. Boundary effects have a significant impact on out-of-plane deformation, wrinkling deformation patterns and wavelengths. But boundary effects have less effect on the out-of-plane deformation of the coated region. The wavelength of wrinkles varies by boundary effects, but the relationship show a "Threshold effect". When $\alpha$ is between 0.55 and 0.65 at a thickness of 0.025 mm, the wavelength reduced sharply, otherwise, the wavelength kept in a steady state. Boundary effects affect the wrinkling pattern in the film region, coated areas get closer to the boundary, stretch-induced folds are more pronounced and the maximum value of out-of-plane deformation increases.

3. Film thickness affects the deformation amplitude in the coating region and the stretching-induced fold deformation extremes in the film region. With little elongation, as the film thickness increases, the deformation in the coating region shows an increasing trend and the wrinkle deformation in the film region shows a decreasing trend. However, bending-induced wrinkling deformation is little affected by the thickness of the film.

4. The dominance of the stretching–bending deformation and the wrinkling deformation was influenced by boundary effects and the thickness of the membrane. As the coated region approaches the boundary, the stretching-induced fold crease deformation in the film region becomes apparent, which is due to the reduction in the stretching-induced buckling load in the film region. As the thickness increases, the bending-induced wrinkling deformation prevails more than the stretching-induced wrinkling deformation (due to increased buckling loads). The boundary shape does not obviously influence the dominance of the deformation type.

5. A new approach to reduce out-of-plane-induced wrinkling deformation is discussed. That is, a partial compensation tension method which reduced the stress in a large film area of a variable stiffness structure, where both stretching- and bending-induced wrinkling deformation would be reduced.

**Author Contributions:** Conceptualization, X.Z., H.L. and X.M.; methodology and software, X.Z.; validation, H.L.; formal analysis, X.Z.; resources, X.M.; writing—original draft preparation, X.Z. and H.L.; writing—review and editing, X.M.; visualization, X.Z.; supervision, X.M. All authors have read and agreed to the published version of the manuscript.

**Funding:** This research was funded by National Natural Science Foundation of China, grant number U20B2033.

**Institutional Review Board Statement:** Not applicable.

**Informed Consent Statement:** Not applicable.

**Data Availability Statement:** Not applicable.

**Conflicts of Interest:** The authors declare no conflict of interest.

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
