# Peer review of "Analysis of Structural Boundary Effects of Copper-Coated Films and Their Application to Space Antennas"

_coatings, doi:10.3390/coatings13091612_

Round 1
Reviewer 1 Report (New Reviewer)
This paper has many issues with a major one being its novelty. In particular, the main contribution of the work is not distinguishable. It seems that various simulations have been conducted using different scenarios. However, this has limited added value. Some additional comments:
1) The language manipulation is bad. Extensive English editing is required. Maybe the novelty can be highlighted more properly if the English are improved.
2) In Figures 4 and 5 there are some results concerning wrinkling, but how is the wrinkling measured? What are the physical units U and U3 in the colormap? Moreover, the "normalized size coated area" makes no sense in the captions. It must be replaced with something more comprehensible.
3) The sentences 150-152 and 178-221 seem to explain the wrinkling mechanism. However, their position in the manuscript makes no sense. A rearrangement must be conducted in section 3.1.
Correspondingly for sentences 306-319.
4) The legend in Figure 8 does not indicate in which physical unit corresponds the curves.
5) The comments for Figures 9 and 10 are not very clear. What are the remarks for these figures?
6) The difference between the models in Fig. 12 are not comprehensible.
The language manipulation is bad. Extensive English editing is required.
Author Response
Please see the attachment.

Reviewer 2 Report (New Reviewer)
20230730 Review Opinions:
General Comments:
This paper analyzes the deformation of the elastic substrate via Abaqus 2020 when it used in a Copper-coated film as a space phased array antenna, and theoretical derivation of force and tension are also derived, which could be useful to some researchers.
However, the results are completely useless for a space phase antenna designer. For antenna designer, the displacement, the rotational angle, and the deformation of the copper patches are relevant, but this paper contains no such information because basically the copper patches are assumed as “fixed” relative to the elastic substrate. If the copper patches position, angle, and form remains the same, then the phase antenna performance is not degraded at all, and the analysis is unnecessary, but this is not true, the position of the patches do change a little under the tension of the substrate.
Nonetheless, I still recommend this paper to be published after revision because the submitted journal is “Coatings” and not “IEEE Trans. on Ant. And Prop.” The methods demonstrated in the paper could be useful for readers of “Coatings”.
Specifics:
l Page 2, line 61: “Qiu et.al [33]” à “Qiu et.al. [33]”
l Fig. 4: What is the unit of u3? Is it in the plane normal direction? If you can describe this in detail, it can actually be modeled in an electromagnetic simulation code.
l Fig. 7: Please define “Wrinkles wavelength”, you are writing a paper about antenna, and wavelength has its special meaning, but I do not know what “Wrinkles wavelength” means.
l Fig. 8: What is the unit in the inset?
l Fig. 9: What is “x coordinate (normalized)”? The deformation is in the plane direction or in the normal direction of the plane? What are the unit in the inset?
l Fig. 10: What is “Normalized patch”?
l Page 11, line 342: “Space-tensioned thin-film phased-array antenna”à “space-tensioned thin-film phased-array antenna”
Minor revision required.
Round 2
Reviewer 1 Report (New Reviewer)
The authors conducted various changes and improved the manuscript significantly. However, there are still some issues:
1) It is mentioned that "large performance changes are possible with small variations in surface accuracy". How is this clarified? This is important since it is linked with the novelty of the paper.
2) The units U and U3 must be described (along with their units) in the materials and methods section as the quantitative deformation factors. It is still not comprehensible what are they corresponding to in terms of wrinkling.
3) Is the definition of the wrinkle wavelength obtained from a well-known citation?
4) The wrinkle wavelength in Figures 4(a) and 4(b) seem to be very different concerning Figure 6(a). However, in Figure 6(b) it seems that this wavelength is equal. Please comment on this. Maybe this is related to the wrinkle wavelength definition.
5) There are many interesting features in Figure 9 that have not been discussed. For example, for the a=0.6 case, the maximum value for the rectangular case is decreased smoothly with the increment of the thickness. On the other hand, the parabolic case shows a more rapid reduction. Some additional features must be discussed.
6) What's the purpose of section 3.3? It seems that there is not any added value from it.
The language manipulation has been improved. However, additional editing is required.
Round 3
Reviewer 1 Report (New Reviewer)
The authors improved significantly their manuscript. Maybe the result description can be somewhat improved, but this is a subjective point of view.
The language manipulation has been improved.
This manuscript is a resubmission of an earlier submission. The following is a list of the peer review reports and author responses from that submission.
Round 1
Reviewer 1 Report
The manuscript details the buckling or wrinkling of a large Kapton foil when deployed using finite element methods. Many of the features discussed are well known in the fracture mechanics community, therefore, it is hard to learn the new information that the manuscript brings. Additionally, the specific reasons WHY the simulations are needed are lacking. The whole time reading the manuscript the question, “why does this matter to the whole device functionality” was not answered. Revisions according to the comments below are highly suggested.
- Why is FEM needed to model these arrays? Does the out-of-plane elastic deformation make a difference to the functionality of the antennas? From what was written, most of the observations were made of the Kapton near the edges where the gripping of the substrate occurs. Any out-of-plane deformation in these areas would not matter to the “patches”.
- There is a difference between wrinkling and buckling when a film and substrate are involved, as in this case. Buckling generally refers to the film delaminating from the substrate and creating a “buckle”, while wrinkling is when the film and substrate elastically deform together. However, the Kapton substrate is mostly modeled and both terms could be used to describe the form of the substrate.
- What is “tension-bending coupling deformation”? This should be well defined in the manuscript.
- The elastic properties of the copper foil are quite high for copper, especially the elastic modulus, which is generally about 120 GPa for untextured copper. How were the values for Kapton and copper determined? What about the yield stresses for both materials? This is also a decisive material parameter that is necessary for FE calculations.
- The elastic deformation at the gripping sites is called “curtaining” or “the towel effect”. It is not new and can be difficult to remove for thin foils. Are the authors trying to remove the effect or only reduce it? Why or why not? Also, if the wavelength of buckling/wrinkling is reduced, that does not automatically indicate that the amplitude of the buckling/wrinkling is also reduced.
- What are the units of color scales presented in all FE figures? The captions should be revised to include more information or the importance of the figure or what is learned from the figures.
- Many of the graph axes are not readable (font too small or needs to be darker) and ALL lines are too thin. Figure 8 is unreadable and the reviewer has no clue what is being presented. Is the deformation plastic or elastic?
- Section 4.1 needs all variables defined. It is unusable in its current form.
- Have the authors considered the viscoelastic properties of Kapton or creep behavior? In orbit, these material properties will be quite important.
Reviewer 2 Report
The paper is a study. There is no novelty or if exists must be clearly clarified. There is a description of the methods and then discussion of parametric simulation results. At 4.1 there is a one page about the nonlinear strain field which is known.
Using a numerical method like the finite element to thin films an error analysis about the approximation accuracy is needed.
Reviewer 3 Report
1. The title should be revised to accurately explain the work and should allude to the theoretical study.
2. It is unclear whether the study is novel.
3. The abstract needs to be enhanced to include the investigation's key results.
4. The keywords should be reduced.
5. The introduction has certain reference mistakes that should be corrected.
6. The introduction is inadequate and should be enhanced to highlight the literature survey that aids the study as well as the significance and uniqueness of the work.
7. The list of references is out of current and does not include recent literature updates. As a result, should be updated, and the data should be evaluated against existing research.
8. Each equation that is utilized should have a link to the original source.
9. All abbreviations should be clarified the first time they appear.
10. The majority of the outdated references should be replaced with fresh ones, as well as the introduction. There are no references in 2023 and only one in 2022.
Round 2
Reviewer 2 Report
The paper remains a good study as very clear indicated at the title. A good engineering work but nothing more. I do not support studies for publication.